

# Impact of spatial proxies on the representation of bottom-up emission inventories: A satellite-based analysis

Guannan Geng[1], Qiang Zhang[1], Randall V. Martin[2,3], Jintai Lin[4], Hong Huo[5], Bo Zheng[6], Siwen Wang[7], and Kebin He[6]

[1]Ministry of Education Key Laboratory for Earth System Modeling, Center for Earth System Science, Tsinghua University, Beijing, China
[2]Department of Physics and Atmospheric Science, Dalhousie University, Halifax, NS, Canada
[3]Harvard-Smithsonian Center for Astrophysics, Cambridge, MA, USA
[4]Laboratory for Climate and Ocean-Atmosphere Studies, Department of Atmospheric and Oceanic Sciences, School of Physics, Peking University, Beijing, China
[5]Institute of Energy, Environment and Economy, Tsinghua University, Beijing, China
[6]State Key Joint Laboratory of Environment Simulation and Pollution Control, School of Environment, Tsinghua University, Beijing, China
[7]Max-Planck-Institute for Chemistry, Mainz, Germany

*Correspondence to*: Q. Zhang (qiangzhang@tsinghua.edu.cn)

**Abstract.** Spatial proxies used in bottom-up emission inventories to derive the spatial distributions of emissions are usually empirical and involve additional levels of uncertainty. Although uncertainties in current emission inventories have been discussed extensively, uncertainties resulting from improper spatial proxies have rarely been evaluated. In this work, we investigate the impact of spatial proxies on the representation of gridded emissions by comparing five gridded $NO_x$ emission datasets over China developed from the same magnitude of emissions and different spatial proxies. GEOS-Chem modeled tropospheric $NO_2$ columns simulated from different gridded emission inventories are compared with satellite-based columns. The results show that differences between modeled and satellite-based $NO_2$ columns are sensitive to the spatial proxies used in the gridded emission inventories. The total population density is less suitable for allocating $NO_x$ emissions than nighttime light data because population density tends to allocate more emissions to rural areas. Determining the exact locations of large emission sources could significantly strengthen the correlation between modeled and observed $NO_2$ columns. When applying industrial gross domestic product (GDP) values and an updated road network map as proxies for the industrial and on-road transport sectors respectively, modeled $NO_2$ columns could better capture pollution hotspots in urban areas and exhibit best performance of the five cases comparing to satellite-based $NO_2$ columns (slope = 1.01 and $R^2$ = 0.85). This analysis provides a framework for information from satellite observations to inform bottom-up inventory development. In the future, more effort should be devoted to the representation of spatial proxies to improve spatial patterns in bottom-up emission inventories.





# 1 Introduction

Emission inventories are essential for predicting spatial and temporal variations in air pollutants and for helping policy makers develop pollution control strategies. The traditional way of developing an emission inventory is the bottom-up approach whereby activity rates and emission factors are aggregated for all known sources (e.g., Streets et al., 2003; Zhang et al., 2009). Emission inventories are most commonly estimated as emission totals of municipal districts (e.g., counties, provinces, or countries) because activity data from statistical yearbooks are typically available for such districts (e.g., Woo et al., 2003; Ohara et al., 2007; Kurokawa et al., 2013). However, gridded emissions are needed to apply inventories in chemical transport models.

Many methods for allocating regional emission totals to grids are available. The most accurate approach involves allocating emissions for which the actual latitude/longitude coordinates of the emitting facilities (e.g., power plants or cement plants) are available. For mobile and area sources (i.e., sources for which the exact emission locations are unknown), parameters of so-called spatial proxies must be used to represent the spatial distributions of emissions. For example, emissions from road transportation sources can be allocated based on road networks, and residential emissions that are strongly related to human activities can be gridded using population densities or nighttime lights (Streets et al., 2003; Woo et al., 2003; Ohara et al., 2007; Zhang et al., 2009; Oda et al., 2011). In many cases, industrial emissions are also allocated by proxies because of the limited information available.

The selection of such spatial proxies is empirical, and their representations of real-world spatial emissions patterns are of considerable concern (Zhou and Gurney, 2011; Andres et al., 2012). Recent efforts have been made to reveal the uncertainties of spatial proxies used in bottom-up $CO_2$ inventories (Rayner et al., 2010; Zhou and Gurney, 2011; Andres et al., 2012; Gately et al., 2013; Gately et al., 2015; Andres et al., 2016), and sophisticated method of allocating emissions to high-resolution grids have been formulated (Gurney et al., 2009; Rayner et al., 2010; Nassar et al., 2013; Asefi-Najafabady et al., 2014). Using population density to downscale fossil fuel emissions can induce biases when analyses are conducted at sub-national scales (Rayner et al., 2010). A high-resolution fossil fuel $CO_2$ emission inventory for the United States further confirms that source heterogeneities are significant and vary by region and sector, indicating that population density is a biased spatial proxy below the state level (Zhou and Gurney, 2011). The above studies suggest that using population density as a spatial proxy may not be appropriate under the hidden assumption that per capita emissions are homogeneous within a region. However, population density remains one of the most widely used spatial proxies in global and regional emission inventories (Zhang et al., 2009; Lu et al., 2011; Kurokawa et al., 2013), and uncertainties transmitted from improper spatial proxies to chemical transport models have rarely been evaluated.

Recent remarkable development in satellite-based remote sensing instruments, or the so-called top-down approach, provides additional constraints to evaluate and improve the existing understanding of emission inventories (Martin et al., 2008; Streets et al., 2013). Tropospheric column densities of important trace gases, such as $NO_2$, $SO_2$, CO, and HCHO, derived from





satellite instruments generate an abundance of useful information on the emission sources of these gases (e.g., Duncan et al., 2010; Boeke et al., 2011; Lin, 2012; Pechony et al., 2013; Stavrakou et al., 2015; Wang et al., 2015; Liu et al., 2016), despite the biases intrinsic to satellite retrievals (Boersma et al., 2008; Lin et al., 2014). Many studies have compared model-simulated column densities with satellite-derived columns to validate the accuracy of bottom-up emissions (e.g., van Noije et

al., 2006; Uno et al., 2007; Kim et al., 2009; Sheel et al., 2010; Lin et al., 2010; Itahashi et al., 2014; Han et al., 2015) and have attributed discrepancies between modeled and satellite-based column densities to errors in the magnitudes and/or spatial distributions of the emission inventories used in their models. Inverse modeling techniques can further derive "top-down" emission inventories with optimized magnitudes and emission spatial distributions (e.g., Martin et al., 2003; Jaeglé et al., 2005; Martin et al., 2006; Wang et al., 2007; Lin, 2012; Tang et al., 2013; Stavrakou et al, 2015). For example, Lin (2012)

found that the widely used Intercontinental Chemical Transport Experiment-Phase B (INTEX-B) inventory may underestimate $NO_x$ emissions in polluted urban areas or near large point sources. Lamsal et al. (2013) demonstrated that urban $NO_2$ pollution is a power law scaling function of population size. The exponent values vary by region, reflecting regional differences in industrial development and per capita emissions. Although these top-down studies have identified uncertainties in the spatial representation of current $NO_x$ inventories and have provided correction factors, these factors are

difficult to incorporate into bottom-up inventories or apply to emissions of other species because less attention has been paid to gridding processes in bottom-up approaches, which vary by sector and are shared across different species.

In this work, we use $NO_x$ as a case to study the influence of spatial proxies on spatial distributions of bottom-up emission inventories because $NO_2$ satellite retrieval is less uncertain and the spatial distributions of tropospheric $NO_2$ columns are similar to those of surface $NO_x$ emissions, especially in the summer when the lifetime of $NO_x$ is short (Richter et al., 2005;

Beirle et al., 2003; Lin, 2012). Based on the same magnitude of $NO_x$ emissions, we develop five sets of gridded emission data using different spatial proxies. We then use these gridded emissions and the nested GEOS-Chem to simulate tropospheric $NO_2$ columns and compare them with satellite-based observations. The effects of spatial proxies on the modeled $NO_2$ columns and representations of different spatial proxies are evaluated and discussed.

## 2 Methods and Data

### 2.1 Review of spatial proxies

We first review the sector-specific spatial proxies utilized in several widely used regional $NO_x$ emission inventories covering China, including TRACE-P (Streets et al., 2003), INTEX-B (Zhang et al., 2009), Regional Emission inventory in Asia (REAS) version 1 (Ohara et al., 2007), and REAS version 2 (Kurokawa et al., 2013), as shown in Table 1. In bottom-up inventories, spatial proxies are usually sector dependent and shared across different species. In general, all of these

inventories rely on similar approaches to allocate emissions from combustion sources. Emissions from large-capacity power



generation units are typically allocated according to their latitude/longitude information, which is the most accurate way to distribute emissions, whereas the population density is used to distribute emissions from small power plants whose locations are unknown (Streets et al., 2003; Ohara et al., 2007; Zhang et al., 2009). Recently, power plant locations from the Carbon Monitoring for Action (CARMA) database (Wheeler and Ummel, 2008) are used to locate emissions (Kurokawa et al., 2013),

providing a larger dataset of power plants than previous work. However, the accuracy of the CARMA database is still uncertain (Oda et al., 2011).

For industrial and residential combustion, population density is the most frequently used spatial proxy because such emissions are believed to be highly correlated with human activities. Total population density is applied for industrial combustion in the TRACE-P and REAS emission inventories (Streets et al., 2003; Ohara et al., 2007; Kurokawa et al., 2013),

which may allocate a large fraction of industrial emissions to rural areas because, in China, the rural population exceeds the urban population (China Statistical Yearbook, National Bureau of Statistics, 2007). In the INTEX-B inventory (Zhang et al., 2009), the urban population is used rather than the total population because industrial activities occur more often in urban areas; however, this strategy is still based on an assumption that per capita industrial emissions are the same across regions within a country. Per capita emissions among regions can differ widely in terms of the regions' levels of economic

development and industrial activity or overall standards of living. Lamsal et al. (2013) found that in different urban areas worldwide, per capita emissions varied significantly because of differing energy consumption rates and energy production infrastructure.

For the transportation sector, road networks are widely used to spatially distribute on-road vehicle emissions based on the assumption that traffic volumes remain the same on different types of roads (e.g., Streets et al., 2003; Ohara et al., 2007).

However, this is not true because of the existence of varied vehicle populations and road capacities. Commonly used road networks in the above inventories are extracted from the Digital Chart of the World (DCW) (DMA, 1993), which has not been updated since 1992. In this case, using DCW road networks to allocate on-road emissions in China may create significant biases in the spatial distributions of emissions because road construction has occurred continuously over the past two decades.

Based on our review of the spatial proxies used in bottom-up emission inventories for China, it can be concluded that such spatial proxies are empirical and may have introduced considerable uncertainties into the spatial distributions of emissions. Studies on fossil fuel $CO_2$ inventories have made tremendous efforts to improve the spatial distribution of emissions based on satellite-derived nighttime light (Oda et al., 2011), fuel sales and traffic data (McDonald et al., 2014), multivariate regressions (Wang et al., 2013; Gately et al., 2015), or combined fossil fuel data assimilation systems (Rayner et al., 2010;

Asefi-Najafabady et al., 2014), shedding new light on ways to improve the spatial distribution of air pollutant inventories.



## 2.2 Gridded NO$_x$ emission inventory

The bottom-up NO$_x$ emission inventory evaluated in this work is obtained from the Multi-resolution Emission Inventory for China (MEIC, http://www.meicmodel.org) for 2006. The MEIC inventory is developed using a technology-based methodology which estimates anthropogenic emissions in China from ~700 emitting sources (Zhang et al., 2007, 2009; Lei

et al., 2011). Table 2 presents the 2006 anthropogenic NO$_x$ emissions estimated by the MEIC for China.

We then develop five sets of gridded NO$_x$ emission data using the same magnitude of emissions from the MEIC and different spatial proxies, as presented in Table 3. Many geographic information system (GIS) grid-based spatial proxies (e.g., population density and road networks) are at a resolution of 1 km $\times$ 1 km. In this work, emissions are first gridded at a resolution of 1 km $\times$ 1 km and then regridded to a resolution of 0.667 $^\circ$lon $\times$ 0.5 $^\circ$lat to fit the GEOS-Chem model. The first

two emission datasets evaluate two types of common spatial proxy maps: population distributions and nighttime lights. In the first gridded emission dataset (S1), all emissions are allocated based on population densities obtained from the Landscan population database (ORNL, 2006). The Gridded Population of the World (GPW) population map is also frequently used to allocate emissions, but we do not include it in our analysis because the uncertainties introduced by differences in population maps are minor (Andres et al., 2016). In the second dataset (S2), all emissions are allocated based on the nighttime lights

map drawn from the Defense Meteorological Satellite Program Operational Linescan System (DMSP-OSL) satellite (http://www.ngdc.noaa.gov/dmsp/download_rad_cal_96-97.html). Figure 1 compares the spatial distributions of the total population and nighttime light data. The nighttime light data present more significant urban-rural gradients than the total population density data and, thus, may better represent differences in economic development levels between urban and rural areas.

Our previous study revealed that the locations of large point sources in emission inventories significantly affect the prediction accuracy of the chemical transport models (Wang et al., 2012). A third dataset (S3) is used to investigate the effects of using the exact locations of emissions from large point sources. S3 is based on S1 but consists of a unit-based power plant emission dataset including the locations of ~6,400 power generation units across China (Liu et al., 2015) to override power sector emissions. The fourth dataset (S4) is based on S3 but uses DCW road networks to allocate on-road

transportation emissions, which have been applied in several widely used regional emission inventories for China.

In the last dataset (S5), two modifications of S4 are made to better represent the spatial patterns of the industrial and on-road transportation sectors. For the industrial sector, we use the Industrial Gross Domestic Product (IGDP) as a first-step spatial proxy instead of population density. Fig. 2 shows the correlations between the normalized industrial emissions and three factors (total population, urban population and IGDP) at the provincial level. IGDP is more closely correlated with emissions

($R^2 = 0.72$) than other factors, indicating that IGDP can better represent the spatial patterns of industrial activities than population density. Using population density as a spatial proxy to allocate industrial emissions assumes that per capita industrial emissions are the same across regions, which may result in underestimations for industrialized regions and



overestimations for rural regions. In S5, provincial emissions are first distributed into counties according to the IGDP of each county, and then county-level emissions are allocated to grids based on the population densities drawn from Landscan.

In S5, we use the county-level vehicle population as a spatial proxy to distribute provincial emissions of on-road transportation to each county. The county-level vehicle population is simulated by the Gompertz Function, which takes into account the effects of populations and income levels on vehicle populations (more information is provided in Zheng et al., 2014). The new proxy overrides the assumption of linear relationships between vehicle ownership and population density or road density. Then, county-level emissions are mapped into grids using GIS-based road networks. As stated in Sect. 2.1, the DCW road networks have not been updated since 1992 and, thus, represent the road conditions of the early 1990s (Fig. 3b). Hence, we update the road network data to a new version (China Digital Road-network Map [CDRM] data developed in 2010 by the National Administration of Surveying, Mapping and Geoinformation of China), which reflects the current road conditions (Fig. 3c). Comparisons between county-level vehicle populations, the total population and these two types of road networks are shown in Fig. 3(d-f). Densely populated regions typically have larger vehicle populations and, therefore, greater road network demands. However, the DCW road network fails to provide detailed accounts of the roads in urban areas, and as a result, the emissions allocated to these regions are underestimated. The updated CDRM road networks can help resolve this shortcoming.

## 2.3 GEOS-Chem model

The GEOS-Chem model is a global, three-dimensional (3-D) model of atmospheric chemistry that includes $> 80$ species and $> 300$ reactions (Bey et al., 2001; Park et al., 2004). The nested-grid GEOS-Chem model developed by Chen et al. (2009) is used in this work. It has a horizontal resolution of $0.667\,°\,$lon $\times\,0.5\,°\,$lat with 47 vertical layers and a nested-grid domain that covers China and most of its neighboring countries ($70\,°$E-$150\,°$E, $11\,°$S-$55\,°$N). The global model, which has a spatial resolution of $2.5\,°\,$lon $\times\,2\,°\,$lat, provides time-varying boundary conditions via the one-way nested approach. Both global and nested simulations are driven by the 3-D meteorological fields of GEOS5 assimilated by the Goddard Earth Observing System (GEOS) at the National Aeronautics and Space Administration (NASA) Global Modeling and Assimilation Office (GMAO; http://gmao.gsfc.nasa.gov/). Mixing in the planetary boundary layer follows a non-local scheme (Lin and McElroy, 2010) that improves upon previous assumptions of a fully mixed boundary layer. Convection occurs according to a modified Relaxed Arakawa-Schubert scheme (Rienecker et al., 2008).

In this study, we use GEOS-Chem version 09-01-02 to simulate tropospheric $NO_2$ columns over China for 2006. The EDGAR emission inventory (Olivier and Berdowski, 2001) is used for global anthropogenic emissions, and the East/Southeast Asia region is replaced with the INTEX-B inventory (Zhang et al., 2009). We further override the anthropogenic $NO_x$ emission inventory for China using the five datasets described in Sect. 2.2. To remove the effects of the initial concentration fields, a 1-year spin up is conducted. We use averaged summer (June, July and August [JJA]) $NO_2$





columns for this evaluation because the short lifetime of $NO_2$ in the summer favors $NO_2$ column linkage with local $NO_x$ emissions. We average daily 2-h modeled tropospheric $NO_2$ columns at a local time of 1300-1500 and sample the model at grids coincident with the daily satellite pixels used in the final average columns.

Grids in the nested-grid GEOS-Chem model are $0.667\,°$ lon $\times 0.5\,°$ lat, and their areas range from 2,500-4,000 km$^2$ over eastern China, comparable to the mean size of a county (~3,000 km$^2$) in this region. Thus, the spatial pattern of emissions evaluated in this work can represent the spatial variations of emissions at the county level. To draw comparisons with other county-level indicators, gridded $NO_2$ column densities simulated using this model are resampled to county averages by area weights. In this work, a total of 2,364 county-level districts are covered, including both counties and municipal districts across China.

## 2.4 Satellite data

The satellite data used in this work comes from the Ozone Monitoring Instrument (OMI) aboard the Aura satellite (Levelt et al., 2006). $NO_2$ slant column densities are derived using a Differential Optical Absorption Spectroscopy (DOAS) algorithm (Platt, 1994; Boersma et al., 2002; Bucsela et al., 2006). The tropospheric slant $NO_2$ column densities used in this work are drawn from the Dutch OMI $NO_2$ (DOMINO) product (version 2, collection 3) (Boersma et al., 2011) available from the Tropospheric Emission Monitoring Internet Service (TEMIS) (http://www.temis.nl/). The air mass factor (AMF) is a multiplicative factor used to convert slant columns into vertical columns (Palmer et al., 2001). The retrieved tropospheric vertical $NO_2$ column is sensitive to the $NO_2$ vertical profile used during the AMF calculation. Following Lamsal et al. (2010), we revise the AMF by replacing the original $NO_2$ profile with that generated from the nested-grid GEOS-Chem model described above. The new $NO_2$ vertical profiles have a finer spatial resolution of $0.667\,°$ lon $\times 0.5\,°$ lat. In this work, we restrict the use of OMI pixels to those at a solar zenith angle of $\leq 70°$ and a cloud fraction of $\leq 0.3$ in the final averaged columns. Pixels at swath edges (five pixels on each side) are rejected to reduce spatial averaging. Finally, each OMI pixel is allocated to $0.667\,°$ lon $\times 0.5\,°$ lat grids by area weights with corner coordinate information to create daily tropospheric vertical $NO_2$ column maps. The retrieved $NO_2$ columns are also resampled from pixels to county averages to draw comparisons with indicators at the county level.

## 3 Results and Discussion

### 3.1 Results

The spatial distributions of tropospheric $NO_2$ columns over China in the summer simulated from five gridded inventories are presented and compared with satellite-observed $NO_2$ columns in Fig. 4. Because all the inventories in model simulations have the same emissions totals, differences among S1-S5 reflect differences in the spatial allocations of the total emissions.



In general, modeled and observed $NO_2$ columns exhibit similar patterns but different fine structures. All modeled cases can reproduce highly polluted areas over the North China Plain and Yangtze River Delta, whereas many pollution hotspots in these regions are underestimated compared to the satellite data in S1. As discussed above, using the total population distribution as a spatial proxy may have misrepresented the urban-rural gradients of economic development levels and allocated disproportionately large fractions of emissions to rural regions. Consequently, urban emission hotspots may have been underestimated because, in China, the rural population exceeds the urban population.

Figure 5 compares modeled and satellite-retrieved tropospheric $NO_2$ columns by county in China for the analyzed five cases. The first column in Fig. 5 compares the model and satellite data for all districts and counties in China for the summer of 2006. Modeled $NO_2$ columns are generally in good agreement with OMI $NO_2$ columns, with regression slopes varying from 0.74~1.01 and $R^2$ values varying from 0.72~0.85. Simulations obtained using S1 substantially underestimate $NO_2$ columns compared to satellite-based columns, especially in densely populated regions. Tropospheric $NO_2$ columns simulated using S2 present more pollution hotspots compared to S1, particularly in economically developed regions, such as the capital cities in each province. Using nighttime lights as a spatial proxy (S2) instead of the total population can improve the model's performance. In this case, the slope and $R^2$ increase to 0.93 and 0.78, respectively, and the normalized mean bias (NMB) increases from -12.8% to -9.1%. These results indicate that the nighttime light map may serve as a better indicator for $NO_x$ emissions than the total population because it can better represent a region's economic development level than the total population.

When using the exact positions of power plants instead of total population density for the power sector (S3 vs. S1), most hotspots in the simulated tropospheric $NO_2$ columns are enhanced, and the discrepancies between the modeled and observed columns are reduced. Model simulations based on S3 correlate better with satellite observations (slope = 0.86 and $R^2$ = 0.83) than those based on S1, proving the importance of determining the positions of large point sources. $NO_2$ columns simulated using S4 have more bias than those generated using S3, mainly because of underestimations of on-road transportation emissions resulting from the use of outdated DCW road networks as a proxy, as discussed above. Model simulations based on S5 agree better with satellite-based $NO_2$ columns than those based on S3, which shows that using IGDP, vehicle population and CDRM road networks can better represent industrial and transportation emissions. Finally, the simulations based on S5 exhibit the best performance for all five cases compared to satellite observations (slope = 1.01 and $R^2$ = 0.85).

To further understand the biases in the model simulations, we divide all counties into three categories—counties in municipalities, urban counties, and suburban counties—and compare the modeled and observed $NO_2$ columns for these three categories (Fig. 5). For the four Chinese municipalities studied, emission totals are allocated from cities to counties; however, for other provinces, emissions are allocated from province to counties. Municipalities are defined as a separate category in the following analysis.



Model underestimation in S1 mainly occurs for urban counties (Fig. 5c, slope = 0.47 and NMB = -23.5%). This approach, which uses the total population as a spatial proxy, assumes that the per capita emissions in different regions in a province are the same. However, because of varied industrial levels and economic patterns, per capita emissions can be very different across regions. According to Lamsal et al. (2013), the tropospheric $NO_2$ column is a power law scaling function of the

population size, and the exponent is affected by regional differences in per capita emissions. Using population density as a spatial proxy in these areas significantly underestimates the emissions in urban areas, as shown in Fig. 5c. Using nighttime lights to allocate emissions can significantly improve the model's performance for urban areas (Fig. 5g, slope = 1.03 and NMB = -3.6%), although overestimations are identified in a few counties. This finding demonstrates the feasibility of using nighttime lights alone as a spatial proxy when more complex indicators are not available. After determining the exact

positions of power plant emissions, the model simulations are substantially improved for all types of regions (Fig. 5j-5l); however, the urban emissions are still underestimated (Slope = 0.67 and NMB = -14.7%), possibly because of underestimations of the industrial and transportation emissions in urban regions when the population density is applied as a spatial proxy. For S4, the model performances are slightly worse than those for S3 for urban regions. In S4, the vehicle populations of different counties are assumed to be linearly correlated with the outdated DCW road networks, and as a result,

the on-road transportation emissions in urban areas may be substantially underestimated. Finally, for S5, the model performances are significantly improved for urban regions, and NMB decreases to 1.8%. Thus, using IGDP and the updated road network as spatial proxies can better represent the emission sources for urban areas.

For the four municipalities studied, total populations and nighttime lights alone cannot effectively represent emission patterns because these proxies are concentrated in city centers, whereas large emission sources, such as power plants and

industrial activities, have largely been relocated away from urban regions in municipalities. When the locations of power emissions are considered, the correlation between the modeled and observed $NO_2$ columns is significantly improved (Fig. 5j, 5n, and 5r). However, using the IGDP and updated road network as spatial proxies disproportionately concentrates emissions in urban areas, resulting in an overestimation of modeled $NO_2$ columns.

Fig. 6 presents the distributions of ratios between simulated and satellite-based county-level $NO_2$ column densities for the

five cases. We remove those counties with OMI $NO_2$ columns of less than $3 \times 10^{15}$ molecules $cm^{-2}$ to avoid the influence of the background areas with more uncertain retrieved columns. After rejecting the background regions, 778 counties (33%) covering much of eastern China remain. Model simulations based on S5 exhibit the best performance. Differences between the modeled and satellite-based $NO_2$ columns are within 20% for 376 counties in S5 compared to 275 and 291 counties in S1 and S2, respectively. Model simulations of S1 present large negative biases compared to satellite-based $NO_2$ columns, with

106 counties underestimated by over 50% (66 counties for S5). However, for S2, positive biases between the modeled and satellite-based $NO_2$ columns exceed 50% for 41 counties (14 counties for S5), indicating that using nighttime light maps may overestimate urban emissions in certain regions.




### 3.2 Uncertainties

This work is subject to several uncertainties. Biases between model simulations and satellite data come not only from emission inventories but also from the model itself or satellite retrievals. Potential errors in nested-grid GEOS-Chem model simulations are compounded by errors in GEOS-5 meteorological fields, PBL heights, and a variety of chemical parameters

selected in a given model. Model simulation errors for eastern China are estimated to present a negative systematic bias of 10–20% (season dependent) plus a random error of 30% according to previous work (Martin et al., 2003; Lin and McElroy, 2011; Lin et al., 2012). Sensitivity simulations (Lin et al., 2012) of the above model factors show that none of them can fully explain the bias between model simulations and satellite observations. Combining all these modifications can achieve better agreement with satellite observations but cannot eliminate the negative biases associated with extremely polluted locations

(Lin et al., 2012), suggesting that model errors are not the primary cause of model-satellite biases for urban areas.

The uncertainties of individual DOMINO v2.0 $NO_2$ column retrievals are estimated at $1.0 \times 10^{15}$ molecules cm$^{-2}$ + 25% (Boersma et al., 2011), which demonstrates the dominance of errors arising during the calculation of AMF in polluted areas (Boersma et al., 2007). In particular, DOMINO v2.0 OMI products do not explicitly account for the effects of aerosols on solar radiation, which are important for the calculation of AMF and particularly significant for eastern China because of its

high aerosol loadings. In Lin et al. (2014), explicitly including aerosol scattering and absorption exerts either positive or negative effects on retrieved $NO_2$, with a mean effect of 14%. However, aerosol effects cannot fully explain the large discrepancies observed between model and satellite results for urban areas.

Other factors, such as the resolution, may also introduce uncertainty. The spatial proxies in S5 are quite good at the resolution of 0.667 ° lon × 0.5 ° lat used in this work, which roughly corresponds to the county level in eastern China.

However, they may not be suitable at other resolutions. Further work based on models with finer grids should be performed to explore appropriate spatial proxies at finer resolutions. Natural emissions from lightning and soil sources are not discussed in this work, although they are suggested to be underestimated by approximately 16% for China for 2006; they account for less than 3% and 6% of anthropogenic emissions, respectively, and even less in highly polluted regions (Lin, 2012).

### 3.3 Discussion

In this work, we use $NO_x$ emissions to relate the biases between model simulations and observations to local emissions and evaluate the impacts of spatial proxies on the distributions of bottom-up emission inventories at the county level using satellite constraints. Insight obtained from this work can be applied to other species generated from fossil fuel combustion (e.g., $SO_2$ and $CO_2$) because they typically come from the same sources. Our method represents a feasible approach to studying species that are difficult to validate directly because suitable observation data are not available and/or lifetimes are

long.



As shown in Sect. 2.1 and described in this work, regardless of how spatial proxies are adjusted in a bottom-up inventory, they are always empirical and contribute uncertainties to the spatial representation of emission inventories. Critical evaluations must be conducted to ensure the accuracy of these proxies. Our work presents a practical means to diagnose this problem and involves using satellite observations as an indicator of ground emissions to determine the relationships between

emissions and local parameters. Our approach also has the attribute of propagating information from satellite observations into bottom-up inventory developments.

The approach presented here can also be expanded to other regions because a universal relationship exists between emissions and the economy. Spatial proxies that work well for China may not be suitable for other regions because of regional differences in energy consumption, industrial development and living standards, as demonstrated in Lamsal et al. (2013).

However, by integrating local satellite observation data, a better understanding of the spatial distribution of emissions and their relationships to local parameters can be obtained, which has implications for the local selection of spatial proxies.

## 4 Concluding Remarks

The spatial proxies used when developing gridded emissions inventories are empirical and can introduce uncertainties in bottom-up emissions inventories. This issue has rarely been evaluated. In this work, we evaluate the effects of spatial proxies

on the representations of spatial distributions of emissions using an integrated framework of bottom-up emission inventories, a chemical transport model, and satellite observations. We first develop five sets of gridded $NO_x$ emissions for China using the same magnitude of emissions from the MEIC and different spatial proxies. The spatial proxies considered in this study include the following: the total population, nighttime lights, the locations of power plants, IGDP, vehicle populations and two different road network datasets. The nested-grid GEOS-Chem model is then used to simulate tropospheric $NO_2$ columns

using the five gridded emissions, and modeled $NO_2$ columns are compared to satellite-based $NO_2$ columns derived from OMI data.

We found that the spatial proxies used in gridded emission inventories significantly affect simulated $NO_2$ columns. The model performance is largely dependent on the representations of urban emissions in the bottom-up inventory, which are very sensitive to spatial proxies. Using the total population density tends to allocate more emissions to rural areas and to

underestimate $NO_2$ columns compared to satellite observations. Nighttime lights represent urban emissions better than population density because they correlate more closely with economic development levels. When using sophisticated combinations of different proxies to represent urban emissions (i.e., positions of large point sources, IGDP, vehicle populations, and the most recent road network), modeled $NO_2$ columns agree better with satellite observations, indicating that improving the spatial representation of emissions could significantly increase the accuracy of emission inventories.





The results of this work emphasize the importance of spatial proxies for bottom-up emission inventory development. Discrepancies between models and observations should be attributed to not only errors in the magnitude of total emission estimates but also spatial proxies. Although the selection of spatial proxies in this work is still empirical and may not represent the best case, we illustrate methods for improving gridded emission inventories by carefully selecting spatial

proxies. This study provides a framework to apply information from satellite observations to inform bottom-up inventory development. The approach used here could be further extended to other species and regions, and more advanced optimized approaches could be introduced into the development of emission inventories of different air pollutants (Asefi-Najafabady et al., 2014; Gately et al., 2013; Nassar et al., 2013). More efforts should be made to improve the spatial distributions of bottom-up emission inventories in the future.

*Acknowledgements*. This work was supported by China's National Basic Research Program (2014CB441301), the National Science Foundation of China (41222036, 41275026, and 21221004), and the MarcoPolo project of the European Union Seventh Framework Programme (FP7/2007-2013) under Grant Agreement number 606953. We acknowledge our free use of tropospheric $NO_2$ column data from OMI: www.temis.nl.

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



**Table 1. Review of the spatial proxies used in regional bottom-up NO$_x$ emission inventories covering China.**

| Inventories | Sectors | Spatial proxies | Data sources |
|---|---|---|---|
| TRACE-P (Streets et al., 2003) | Large power plants | Location | RAINS-Asia (Shah et al., 2000) and GEIA inventory (Graedel et al., 1993) |
| | Small power plants | Total population | LandScan[1] |
| | Industrial combustion | Total population | LandScan |
| | Residential fossil fuel | Total population | LandScan |
| | Residential biofuel | Rural population | LandScan |
| | On-road transport | Road networks | DCW[2] |
| | Off-road transport | Total population | LandScan |
| REAS v.1.1 (Ohara et al., 2007) | Large power plants | Location | China State Grid Company |
| | Small power plants | Total population | LandScan |
| | Industrial combustion | Total population | LandScan |
| | Residential fossil fuel | Total population | LandScan |
| | Residential biofuel | Rural population | LandScan |
| | On-road transport | Road networks | DCW |
| | Off-road transport | Total population | LandScan |
| REAS v.2 (Kurokawa et al., 2013) | Large power plants | Location | CARMA (Wheeler and Ummel, 2008) |
| | Small power plants | Total population | GPWv3[3] |
| | Industrial combustion | Total population | GPWv3 |
| | Residential fossil fuel | Total population | GPWv3 |
| | Residential biofuel | Rural population | GPWv3 and GRUMPv1[4] |
| | On-road transport | Road networks | DCW |
| | Off-road transport | Total population | GPWv3 |
| INTEX-B (Zhang et al., 2009) | Large power plants | Location | Ministry of Environmental Protection |
| | Small power plants | Total population | LandScan |
| | Industrial combustion | Urban/rural population | LandScan |
| | Residential fossil fuel | Total population | LandScan |
| | Residential biofuel | Rural population | LandScan |
| | On-road transport | Road networks | DCW |
| | Off-road transport | Total population | LandScan |

[1]LandScan Global Population database (ORNL, 1999, 2001, 2006)

[2]DCW, Digital Chart of the World (DMA, 1993)

[3]GPWv3, Gridded Population on the World (CIESIN et al., 2005, 2011)

5 [4]GRUMPv1, Global Rural-Urban Mapping Project (CIESIN et al., 2005, 2011)



**Table 2. Anthropogenic NOₓ emissions by sector in China for 2006 from the MEIC inventory.**

| Sector | Annual Emissions (Tg) |
|---|---|
| Power plants | 8.31 |
| Industry | 7.34 |
| Transport | 6.39 |
| Residential | 1.00 |
| Total | 23.04 |




**Table 3. Spatial proxies used in the gridding process for the five emission scenarios developed in this study.**

| Gridding process | Sectors | S1 | S2 | S3 | S4 | S5 |
|---|---|---|---|---|---|---|
| **Province to county** | Power Plant | TP[a] | NL[b] | N/A | N/A | N/A |
| | Industry | TP | NL | TP | TP | IGDP[c] |
| | Residential | TP | NL | TP | TP | TP |
| | On-road transport | TP | NL | TP | DCW[d] | VP[e] |
| | Off-road transport | TP | NL | TP | TP | TP |
| **County to grid** | Power Plant | TP | NL | PS[f] | PS | PS |
| | Industry | TP | NL | TP | TP | TP |
| | Residential | TP | NL | TP | TP | TP |
| | On-road transport | TP | NL | TP | DCW | CDRM[g] |
| | Off-road transport | TP | NL | TP | TP | TP |

[a]TP: Total population from the Landscan population database (ORNL, 2006);

[b]NL: Nighttime light from the DMSP-OSL satellite (http://www.ngdc.noaa.gov/dmsp/download_rad_cal_96-97.html);

[c]IGDP: Industrial GDP (China Statistical Yearbook, National Bureau of Statistics, 2007);

5    [d]DCW: Road networks from the Digital chart of the world (DMA, 1993);

[e]VP: Vehicle population (Zheng et al., 2014);

[f]PS: Coordinates of point sources (Liu et al., 2015);

[g]CDRM: Road networks from the China Digital Road Map developed by the National Administration of Surveying, Mapping and Geoinformation of China.





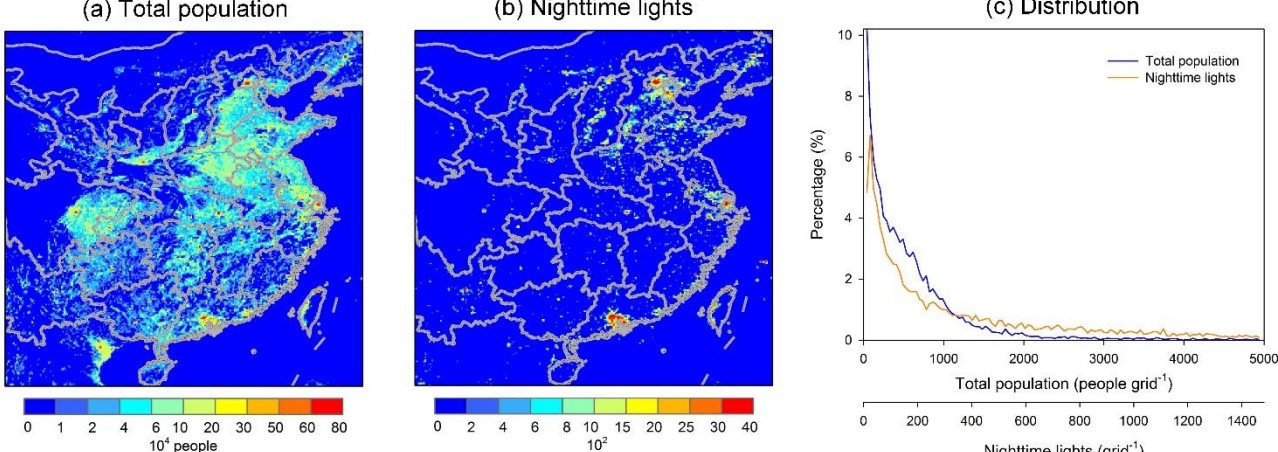

Fig. 1. Spatial patterns of (a) the total population density and (b) nighttime lights for eastern China, and (c) the distributions of total population density and nighttime lights in China at a resolution of $0.1° \times 0.1°$.





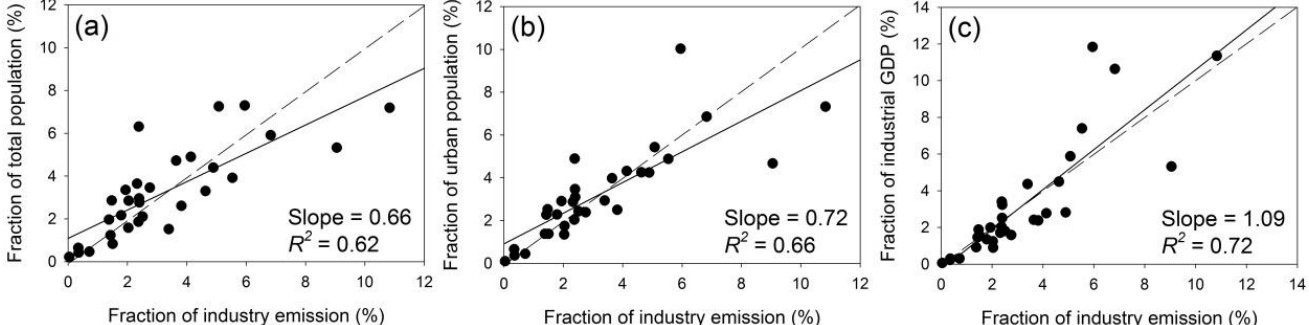

**Fig. 2. Correlations between normalized provincial industrial emissions and three types of spatial proxies: (a) total population, (b) urban population and (c) IGDP data.**





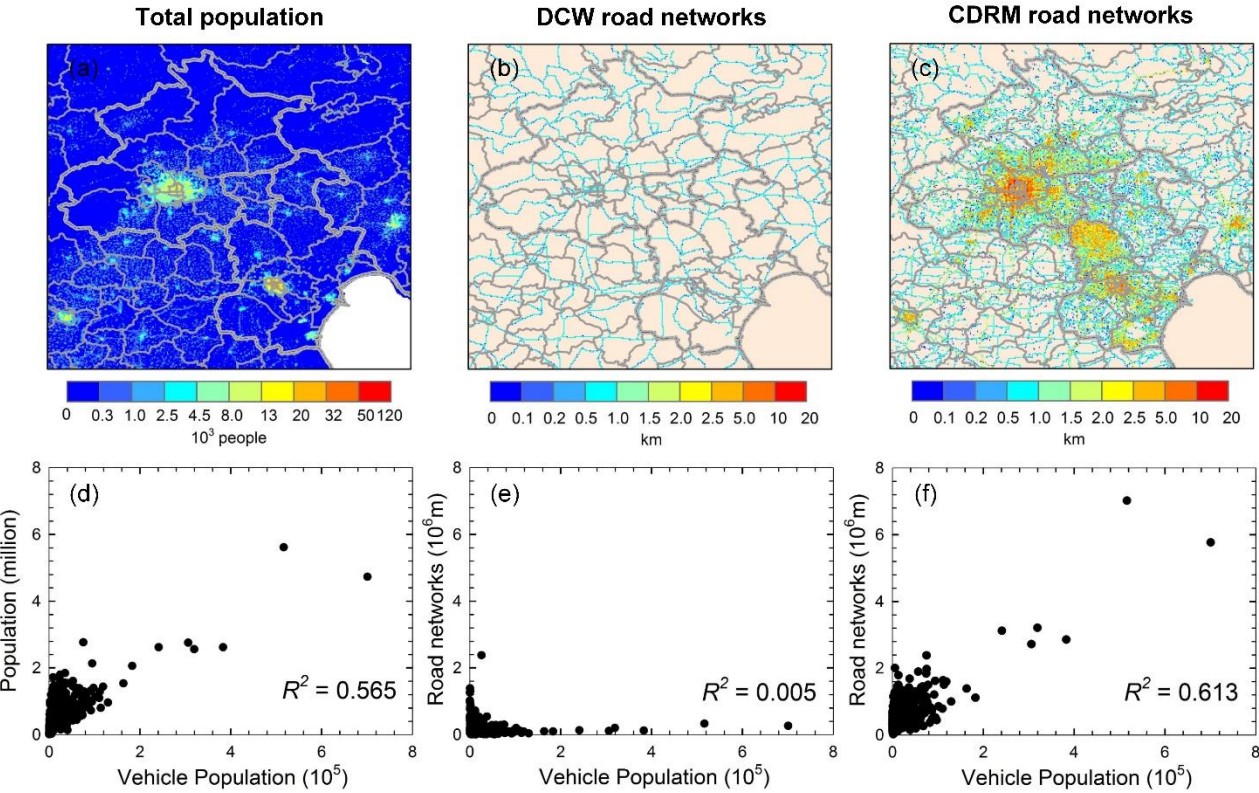

Fig. 3. Comparisons between the total population (left column), DCW road networks (middle column) and CDRM road networks (right column). (a-c) Maps of the total population and two road networks in the Beijing-Tianjin region as an example. (d-f) Comparison with county-level vehicle populations.




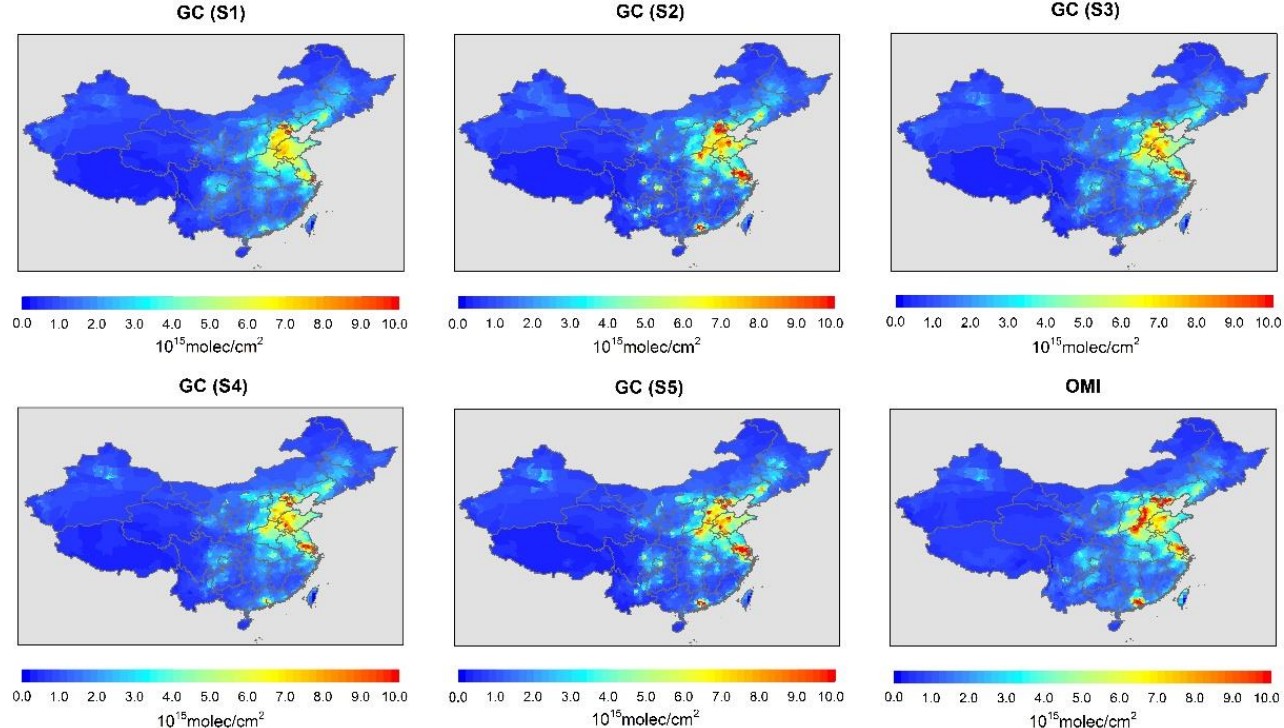

**Fig. 4. Spatial distributions of summer averaged tropospheric NO₂ columns modeled by GEOS-Chem based on S1-S5 emissions and retrieved from the OMI for 2006.**



**Fig. 5.** Comparisons between county-level model simulations and the five emission cases and OMI NO₂ columns for all counties (first column), urban districts in municipalities (second column), urban districts in other cities (third column) and other counties (fourth column). The color of each symbol corresponds to the population density in the county specified by that symbol. The dotted line has a slope of 1.





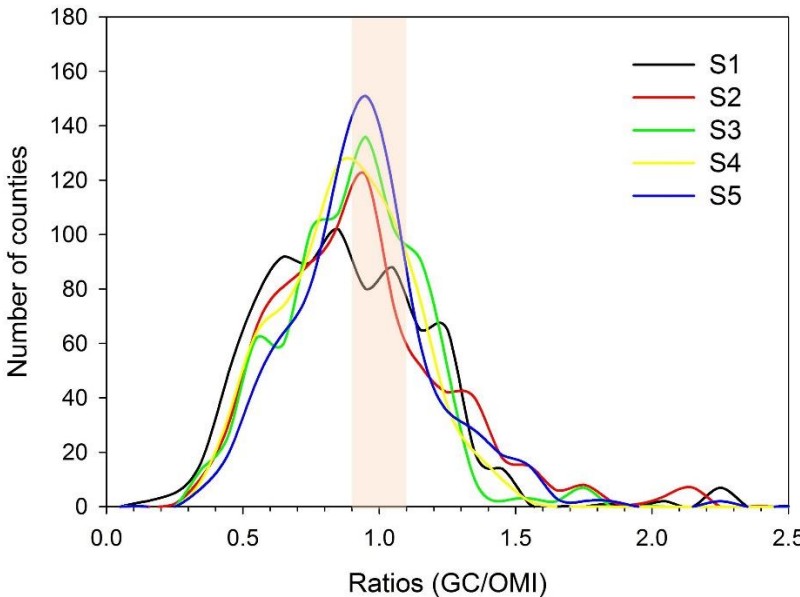

**Fig. 6. Distributions of the ratios between the county-level model simulations and satellite observations from the five emission inventories. The shaded region indicates a range of 0.9~1.1.**