# Peer review of "Impact of spatial proxies on the representation of bottom-up emission inventories: A satellite-based analysis"

_Atmospheric Chemistry and Physics, 2016_

## Referee Comment (RC1) · Anonymous Referee #1 · 29 Nov 2016

Title: Impact of spatial proxies on the representation of bottom-up emission inventories: a satellite-based analysis Author(s): Guannan Geng et al. MS No.: acp-2016-905

General Comments: The authors explore the sensitivity of NOx VCD to spatial distribution of NOx emissions. Generally, the manuscript presents a sound analysis of the influence of the selected spatial surrogates. It is well-written with clear connections between the objectives and results. There are a few key places the analysis could be improved or should at least be addressed in the narrative. (1) The review of spatial surrogates used by other studies raises questions about the sectors and scenarios used in this study. (2) Further, the scenarios applied do not distinguish between critical sector-based allocation uncertainties. (3) The spatial allocation to the road network is

not sufficientlly discussed. (4) The impact of the prior vertical distribution should be discussed further with respect to artifacts introduced by increasing or decreasing urban/rural gradients in the AMF calculation. (5) The AMF application needs to be more clearly connected to results in Fig 4-6. (6) The OMI NO2 threshold appears to be applied for only some of the analyses (Fig 6) – to what extent would this change bias calculations in previous sections. (7) Lastly, how would you expect the systematic 10% low-bias to affect the outcome of this study? More discussion of each point is provided below.

(1) The review of spatial surrogates shows that previous studies have use more sectors than the MEIC and use better spatial allocation than S1-S3. For example, none of the reviewed inventories uses total population to allocate large power plants, on-road transportation, or biofuels. This raises questions about the reasonability of S1 and S3. The authors should clearly state the value of including "non standard-of-practice" references in the evaluation. The authors should also specifically address the lack of residential biofuel and Table 2 should include a distinction between off-road vs on-road transportation emissions. The S4 and S5 simulations seem most consistent with previous work, and should be the focus of the paper's contribution. Particularly in the conclusions, it should be noted that what the standard of practice is, so that the results are not incorrectly extrapolated.

(2) The five scenarios do not distinguish between the uncertainty in transportation and industrial spatial allocation. The authors correctly point out that the DCW road network is outdated and may be an important source of uncertainty. They then do not quantify the effect of the road network, instead they combine industrial and road network allocations (S4 to S5). Especially because these changes are most relevant to the reviewed inventories (Table 1), the effect of each uncertainty would be very valuable. Can the authors add a simulation or infer contributions from existing data?

(3) The description of the road network allocation lacks specificity and is inconsistent with other parts of the manuscript. The manuscript simply says that the S4 "uses DCW

road networks to allocate on-road transportation sectors." First, Table 2 does not distinguish between on-road and off-road so the effect of allocation is not clear. Second, spatial allocation to a road network can benefit from distinctions between road classes and capacities. The manuscript does not discuss how/if primary or secondary roads are treated differently. Improving the characterization of this process would improve the clarity and reproducibility of results.

(4) The manuscript explains the use of the GC model results as a prior, but does not distinguish between the role of the prior and the change in simulated VCD. Recent work has suggested that higher resolution priors improve the model bias. This work does not distinguish between the role of the prior and the change in simulated concentration. Can the authors clarify the relative roles?

(5) The discussion of AMF in section 2.4 is difficult to connect to results in Figures 4-6. If the AMF is dependent on the GC profile, then the AMF should be scenario specific (S1-S5). In the cited Lamsal et al. (2010), the retrieval was modified to to create a satellite product consistent with the GEOS-Chem run (DP_GC). In this manuscript, Pg 7 ln 18-19 says that the AMF was revised using the profile "from the nested GC model described above," which could be interpreted as inclusive of S1-S5. Figure 4, however, has only one OMI product panel. In Fig 5 and 6, the individual scenarios may be compared to scenario-specific VCD. Please clarify.

(6) The low OMI NO2 values are removed from the PDF in Fig 6 because according to the authors their priors are uncertain. To what extent was the analysis characterized in Fig 5 influenced by grid cells with uncertain priors? The "other" counties, due to the large number, appear to dominate the NMB calculations. The "other" counties are more frequently below 3e15, so the sensitivity to this threshold seems relevant. From the scatter plots, I doubt it will affect your conclusions. Please discuss the reason between using different results.

(7) Evaluation of trends is common with OMI due to the known biases. For this work,

the bias is important to the evaluation of the model. The uncertainties section (3.2) addresses some the uncertainties. Specifically, how does the source of the systematic low-bias (pg 10, 6) possibly affect the outcome of the study? If the satellite retrievals are high-biased, how might the conclusions have been different?

Specific Comments:

- general, 4 municipalities are discussed, but never defined. People not familiar with the region won't know what they are. - general, use specific vertical or slant column terminology throughout. (e.g., section 3.1 Results; Fig 4 caption; etc) - pg 2, 27 - This statement seems to suggest a previous over-reliance on population and, yet, the studies cited here all use other spatial surrogates including road networks, and GDP. Zhang et al. (2009) use Streets 2003, which in Table 1 and the paper uses more than population density as a surrogate. Lu et al. (2011) uses road networks and GDP. Further, this paper does not evaluate the errors introduced by using population for residential or sub-province industry. - pg 5, 28-29 - Will correlation at the provincial level be a good surrogate for sub-province (i.e., county) allocation? Or could emissions be more related to sector than IGDP? Still, seems a good improvement. - pg 6, 4 - Details on the Gompertz formulation would be appropriate here because the allocation of transportation emissions becomes a central outcome. - pg 6, circa 4 - Details on the road network allocation are critical. Are all roads weighted by length density (km/km2) or is road capacity addressed (primary vs secondary roads)? - pg 6, 28 - Which version of EDGAR was used? - pg 7, 19 - The new product resolution is unclear to me. Can you clarify the original resolution and the new resolution? - pg 7, 16-21 - Am I correctly interpreting that OMI NO2 VCD in Figure 5 uses a AMF consistent with that emission inventory? - pg 8, 27 - more details of how the counties are assigned would be useful. - pg 10, 11 - Rephrase. It is not clear which error you are referring to. - pg 11, 6-7 - This is most interesting for the S4 vs S5 differences. The use of newer road networks seems like an obvious best-practice. The IGDP seems like a less obvious improvement, which is why seeing the contribution of each would be nice.

Technical Corrections:

- pg 1, 26 - use IGDP here since that is how it is subsequently used or remove acronym, which is not used in the abstract or elsewhere without redefinition. - Table 1 - citations in table should, but do not appear in bibliography. - Fig 6 - How did your PDF bandwidth affect the shape of S1 and the tails of S2-S3?

—————————————————

---

## Referee Comment (RC2) · P. Rayner (Referee) · 5 Jan 2017

This paper compares several methods for "downscaling" CO2 emissions using different spatial proxies. It compares the proxies by comparison with satellite NO2 column measurements which are regarded as proxies for anthropogenic CO2 concentrations. The paper is clearly in scope for ACP. It is well written with extensive and useful citation of the relevant literature.

I find it to be a very good paper. Its aim is limited, pointing out a sensitivity to the spatial proxies used and making a suggestion about the best choice, at least for China. It achieves this aim well however. Interestingly the choice is similar for the U.S. where the validation dataset is the emission inventory from the VULCAN project.

[Figure]

My only concern for the paper is lack of availability of the various underlying datasets. I'm not sure what the current policy of Copernicus journals is but verification of this analysis plus more general utility requires the community to be able to access the various emission products used in the paper. I believe availability of these datasets should be a precondition for the publication of the article but this is a question for the journal itself.

---

## Author Comment (AC1) · 7 Feb 2017

This paper compares several methods for "downscaling" $CO_2$ emissions using different spatial proxies. It compares the proxies by comparison with satellite $NO_2$ column measurements which are regarded as proxies for anthropogenic $CO_2$ concentrations. The paper is clearly in scope for ACP. It is well written with extensive and useful citation of the relevant literature. I find it to be a very good paper. Its aim is limited, pointing out a sensitivity to the spatial proxies used and making a suggestion about the best choice, at least for China. It achieves this aim well however. Interestingly the choice is similar for the U.S. where the validation dataset is the emission inventory from the VULCAN project.

**Response:** We thank Referee #2 for the encouraging comments.

My only concern for the paper is lack of availability of the various underlying datasets. I'm not sure what the current policy of Copernicus journals is but verification of this analysis plus more general utility requires the community to be able to access the various emission products used in the paper. I believe availability of these datasets should be a precondition for the publication of the article but this is a question for the journal itself.

**Response:** Spatial proxies from the best case of the six scenarios in this study are utilized to allocate emissions in the MEIC model. Gridded emissions from the MEIC model are publicly available via http://www.meicmodel.org/.

---

## Author Comment (AC2) · 7 Feb 2017

**Anonymous Referee #1:**

General Comments: The authors explore the sensitivity of NOx VCD to spatial distribution of NOx emissions. Generally, the manuscript presents a sound analysis of the influence of the selected spatial surrogates. It is well-written with clear connections between the objectives and results. There are a few key places the analysis could be improved or should at least be addressed in the narrative. (1) The review of spatial surrogates used by other studies raises questions about the sectors and scenarios used in this study. (2) Further, the scenarios applied do not distinguish between critical sector-based allocation uncertainties. (3) The spatial allocation to the road network is not sufficiently discussed. (4) The impact of the prior vertical distribution should be discussed further with respect to artifacts introduced by increasing or decreasing urban/rural gradients in the AMF calculation. (5) The AMF application needs to be more clearly connected to results in Fig 4-6. (6) The OMI NO2 threshold appears to be applied for only some of the analyses (Fig 6) – to what extent would this change bias calculations in previous sections. (7) Lastly, how would you expect the systematic 10% low-bias to affect the outcome of this study? More discussion of each point is provided below.

**Response:** We thank Referee #1 for the encouragement and for the valuable comments to improve our manuscript. Responses to each point are addressed as below.

(1) The review of spatial surrogates shows that previous studies have use more sectors than the MEIC and use better spatial allocation than S1-S3. For example, none of the reviewed inventories uses total population to allocate large power plants, on-road transportation, or biofuels. This raises questions about the reasonability of S1 and S3. The authors should clearly state the value of including "non standard-of-practice" references in the evaluation. The authors should also specifically address the lack of residential biofuel and Table 2 should include a distinction between off-road vs onroad transportation emissions. The S4 and S5 simulations seem most consistent with previous work, and should be the focus of the paper's contribution. Particularly in the conclusions, it should be noted that what the standard of practice is, so that the results are not incorrectly extrapolated.

**Response:** Population density is a widely used spatial proxy in bottom-up emission inventories for different sources, including small power plants (Streets et al., 2003; Zhang et al., 2009), industry (Zhang et al., 2009), transportation (Woo et al., 2003) and residential (Kurokawa et al., 2013). Here we use population density as the base case and make modifications upon it to evaluate different spatial proxies used in different sectors.

Emissions from residential biofuel use were included in this work and aggregated to "residential" sector. We have clarified this in the revised manuscript. Table 2 is revised to separate on-road and off-road transportation emissions.

(2) The five scenarios do not distinguish between the uncertainty in transportation and industrial spatial allocation. The authors correctly point out that the DCW road network is outdated and may be an important source of uncertainty. They then do not quantify the effect of the road network, instead they combine industrial and road network allocations (S4 to S5). Especially because these changes are most relevant to the reviewed inventories (Table 1), the effect of each uncertainty would be very valuable. Can the authors add a simulation or infer contributions from existing data?

**Response:** We thank the Referee's suggestion to improve our manuscript. We have added a new scenario to separate the effect of transportation and industrial sectors. The new scenario is defined as the new S5 and the original S5 is now S6 (see revised Table 3). The new S5 is based on S4 but uses county-level vehicle population as a spatial proxy to distribute provincial emissions of on-road transportation to each county. Then the county-level emissions are mapped into grids using the updated CDRM road networks.

Model simulations based on S5 agree better with satellite-based $NO_2$ columns (slope = 0.95 and $R^2$ = 0.86) than those based on S3 and S4, which shows that using vehicle population and CDRM road networks can better represent transportation emissions.

We have revised the manuscript to include the descriptions of the newly added scenario.

(3) The description of the road network allocation lacks specificity and is inconsistent with other parts of the manuscript. The manuscript simply says that the S4 "uses DCW road networks to allocate on-road transportation sectors." First, Table 2 does not distinguish between on-road and off-road so the effect of allocation is not clear. Second, spatial allocation to a road network can benefit from distinctions between road classes and capacities. The manuscript does not discuss how/if primary or secondary roads are treated differently. Improving the characterization of this process would improve the clarity and reproducibility of results.

**Response:** We thank the Referee for the suggestions. We added more information about the allocation process for transportation sector in the revised manuscript.

First, we revised Table 2 to provide the magnitude of $NO_x$ emissions for on-road and off-road sectors. On-road emissions were 4.51 Tg over China in 2006, 2.4 times larger than that of off-road emissions (1.88 Tg).

Second, in S4, on-road transportation emissions are allocated according to the road's length, which is the common practice in previous studies. We have revised the sentence to clarify this:

"The fourth dataset (S4) is based on S3 but uses DCW road networks to allocate on-road transportation emissions, which have been applied in several widely used regional emission inventories for China. The emissions are allocated according to the road's length, neglecting the distinctions of road classes and capacities".

In the improved method proposed in S5, we have considered the difference between different road types by using total vehicle kilometers traveled data as allocation weights. We now specify in the sentence that: "Then, county-level emissions are mapped into grids using GIS-based road networks, which includes various road types (highway, national, provincial and county roads) and the total vehicle kilometers traveled data is used as allocation weights on different road types."

(4) The manuscript explains the use of the GC model results as a prior, but does not distinguish between the role of the prior and the change in simulated VCD. Recent work has suggested that higher resolution priors improve the model bias. This work does not distinguish between the role of the prior and the change in simulated concentration. Can the authors clarify the relative roles?

**Response:** In the revised manuscript, we used the $NO_2$ products adjusted by each scenario's vertical profile for comparison with corresponding model results. As pointed out by the referee, we found that higher resolution prior profiles could reduce the bias by 3%-6% between modeled and satellite data for different scenarios. We have added this in the revised manuscript.

(5) The discussion of AMF in section 2.4 is difficult to connect to results in Figures 4-6. If the AMF is dependent on the GC profile, then the AMF should be scenario specific (S1-S5). In the cited Lamsal et al. (2010), the retrieval was modified to create a satellite product consistent with the GEOS-Chem run (DP_GC). In this manuscript, Pg 7 ln 18-19 says that the AMF was revised using the profile "from the nested GC model described above," which could be interpreted as inclusive of S1-S5. Figure 4, however, has only one OMI product panel. In Fig 5 and 6, the individual scenarios may be compared to scenario-specific VCD. Please clarify.

**Response:** We thank the Referee for pointing out this issue. We now used scenario-specific OMI $NO_2$ data during comparisons. The changes caused by different $NO_2$ products is minor and do not affect our conclusions.

Description for satellite data in Section 2.4 is revised as: "Following Lamsal et al. (2010), we revise the AMF by replacing the original $NO_2$ profile with that generated from the nested-grid GEOS-Chem model using S1-S6 emissions described above. The new $NO_2$ vertical profiles have a finer spatial resolution of 0.667° lon × 0.5° lat compared to the original resolution of 3°

lon × 2° lat. Six scenario-specific OMI $NO_2$ products are generated for comparison with corresponding model results, and higher resolution prior profiles could reduce the bias by 3%-6% between modeled and satellite data for different scenarios."

(6) The low OMI $NO_2$ values are removed from the PDF in Fig 6 because according to the authors their priors are uncertain. To what extent was the analysis characterized in Fig 5 influenced by grid cells with uncertain priors? The "other" counties, due to the large number, appear to dominate the NMB calculations. The "other" counties are more frequently below 3e15, so the sensitivity to this threshold seems relevant. From the scatter plots, I doubt it will affect your conclusions. Please discuss the reason between using different results.

**Response:** We conducted comparisons in Figure 5 with and without data in the background area (i.e., counties with $NO_2$ columns below 3e15 molec/cm$^2$) for "all districts and counties", and especially for "other counties". The results are shown as below:

| All counties | Including background area | | | | | | Excluding background area | | | | | |
|---|---|---|---|---|---|---|---|---|---|---|---|---|
| | S1 | S2 | S3 | S4 | S5 | S6 | S1 | S2 | S3 | S4 | S5 | S6 |
| Slope | 0.78 | 0.94 | 0.87 | 0.85 | 0.95 | 1.01 | 0.62 | 0.88 | 0.78 | 0.78 | 0.95 | 1.07 |
| $R^2$ | 0.75 | 0.81 | 0.83 | 0.83 | 0.86 | 0.85 | 0.40 | 0.57 | 0.58 | 0.58 | 0.67 | 0.69 |
| NMB | -11.1 | -6.0 | -9.3 | -10.2 | -6.5 | -5.5 | -13.0 | -4.3 | -9.4 | -11.0 | -5.5 | -3.4 |
| RMSE | 1.30 | 1.20 | 1.08 | 1.10 | 1.02 | 1.11 | 2.20 | 2.00 | 1.83 | 1.85 | 1.71 | 1.85 |

| Other counties | Including background area | | | | | | Excluding background area | | | | | |
|---|---|---|---|---|---|---|---|---|---|---|---|---|
| | S1 | S2 | S3 | S4 | S5 | S6 | S1 | S2 | S3 | S4 | S5 | S6 |
| Slope | 0.78 | 0.88 | 0.84 | 0.82 | 0.87 | 0.90 | 0.54 | 0.71 | 0.71 | 0.70 | 0.76 | 0.83 |
| $R^2$ | 0.80 | 0.81 | 0.85 | 0.84 | 0.86 | 0.86 | 0.38 | 0.44 | 0.55 | 0.54 | 0.58 | 0.60 |
| NMB | -11.4 | -9.1 | -10.6 | -11.4 | -9.4 | -9.7 | -13.7 | -8.0 | -11.2 | -12.6 | -9.2 | -8.7 |
| RMSE | 1.02 | 1.02 | 0.89 | 0.91 | 0.87 | 0.88 | 1.81 | 1.78 | 1.57 | 1.61 | 1.31 | 1.54 |

The correlation slope, $R^2$, NMB and RMSE values are changed when excluding counties in the background area; however, it does not affect our conclusions about the improvements among different scenarios. Therefore we feel it is better to keep all the counties in our analysis.

In Figure 6, we examine the distributions of the ratios between the modeled and satellite $NO_2$ vertical columns. Ratios can be strongly affected when the satellite $NO_2$ value is low and with large uncertainties, therefore we have to exclude the background areas during this analysis.

(7) Evaluation of trends is common with OMI due to the known biases. For this work, the bias is important to the evaluation of the model. The uncertainties section (3.2) addresses some the uncertainties. Specifically, how does the source of the systematic low-bias (pg 10, 6) possibly affect the outcome of the study? If the satellite retrievals are high-biased, how might the conclusions have been different?

**Response:** Biases in the satellite $NO_2$ data could affect the comparisons for different scenarios. Previous studies indicated that the OMI $NO_2$ columns could be biased high for 20% over China (Irie et al., 2008; Lin et al., 2014). In this case, the best scenario of spatial proxies would be S4 instead of S6. If the OMI data is biased low instead of biased high, the best scenario will remain the same but with less agreement compared to satellite observations. However, given the high sensitivity of modeled $NO_2$ columns to spatial proxies, we can still conclude that the spatial proxies used in gridded emission inventories could affect the representation of bottom-up emission inventory significantly.

The above discussions have been added to the Sect. 3.2 of the revised manuscript.

Specific Comments:
- general, 4 municipalities are discussed, but never defined. People not familiar with the region won't know what they are.

**Response:** The four municipalities in China are Beijing, Tianjin, Shanghai and Chongqing. We have added the name of the cities in the revised manuscript.

- general, use specific vertical or slant column terminology throughout. (e.g., section 3.1 Results; Fig 4 caption; etc)

**Response:** Tropospheric $NO_2$ vertical column densities were used in this study. We have revised the manuscript as suggested.

- pg 2, 27 - This statement seems to suggest a previous over-reliance on population and, yet, the studies cited here all use other spatial surrogates including road networks, and GDP. Zhang et al. (2009) use Streets 2003, which in Table 1 and the paper uses more than population density as a surrogate. Lu et al. (2011) uses road networks and GDP. Further, this paper does not evaluate the errors introduced by using population for residential or sub-province industry.

**Response:** Population density is a widely used spatial proxy in bottom-up emission inventories for different sources, including small power plants (Streets et al., 2003; Zhang et al., 2009), industry (Zhang et al., 2009), transportation (Woo et al., 2003) and residential

(Kurokawa et al., 2013). In this work, we use population density as the base case and make modifications upon it to evaluate different spatial proxies used in different sectors.

- pg 5, 28-29 - Will correlation at the provincial level be a good surrogate for sub-province (i.e., county) allocation? Or could emissions be more related to sector than IGDP? Still, seems a good improvement.

**Response:** Figure 2 (Figure 3 for now) is used to show the significant correlation between industrial emissions and IGDP. We use provincial data for this comparison because emissions in the MEIC model are estimated at province-level. Based on this relationship, we allocate emissions from province to counties using IGDP of each county.

We have revised the manuscript as: "Fig. 3 shows the correlations between the normalized industrial emissions and three factors (total population, urban population and IGDP) at the provincial level. We use provincial data for this comparison because emissions in the MEIC model are estimated at province-level."

- pg 6, 4 - Details on the Gompertz formulation would be appropriate here because the allocation of transportation emissions becomes a central outcome.

**Response:** The vehicle growth of a region is highly correlated to its economic development (e.g., per-capita GDP), and the Gompertz function (an S-shaped curve with three phases of slow, fast and, finally, saturated growth) is often used to establish the relationship between per-capita GDP and total vehicle ownership. In this study, we used the Gompertz function to hindcast total vehicle ownership at county level using historical GDP data:

$$\text{Gompertz Function: } V = V^* \times e^{\alpha e^{\beta E}},$$

where $V$ represents total vehicle ownership (vehicles/1000 people); $V^*$ represents the saturation level of total vehicle ownership (vehicles/1000 people); E represents per-capita GDP; and $\alpha$ and $\beta$ are two negative parameters that determine the shape of the curve.

We have added more descriptions of the Gompertz Function in the manuscript: "We use the county-level vehicle population as a spatial proxy to distribute provincial emissions of on-road transportation to each county (an administrative unit one level lower than city). The county-level vehicle population is simulated by the Gompertz Function ($V = V^* \times e^{\alpha e^{\beta E}}$), where $V$ and $V^*$ represents actual and saturation level of total vehicle ownership (vehicles/1000 people) separately and $E$ represents per-capita GDP (more information is provided in Zheng et al., 2014)."

- pg 6, circa 4 - Details on the road network allocation are critical. Are all roads weighted by length density (km/km$^2$) or is road capacity addressed (primary vs secondary roads)?

**Response:** The spatial allocation of vehicle emissions was processed in two steps. First, we used the total vehicle kilometers traveled allocation weights on different types of roads (highway, national, provincial and county roads) to split vehicle activity. Second, we divided the county-level emissions according to road type, then plotted the results based on road density for hot-stabilized emissions and urban population distributions for start and evaporation emissions.

We revised the sentence to provide more information about the allocation process: "Then, county-level emissions are mapped into grids using GIS-based road networks, which includes various road types (highway, national, provincial and county roads) and the total vehicle kilometers traveled data is used as allocation weights on different road types".

**Response:** We used GEOS-Chem version 09-01-02 in this study and the default global emissions are EDGAR v3.

- pg 7, 19 - The new product resolution is unclear to me. Can you clarify the original resolution and the new resolution?

**Response:** The original NO$_2$ vertical profile used during the AMF calculation in the DOMINO v2 product is adopted from the TM4 model, which has a spatial resolution of 3° lon × 2° lat. In this work, we used the new NO$_2$ vertical profile taken from the nested-grid GEOS-Chem model, which has a finer spatial resolution of 0.667° lon × 0.5° lat. We have revised the sentence to more clearly state the issue: "The new NO$_2$ vertical profiles have a finer spatial resolution of 0.667° lon × 0.5° lat compared to the original resolution of 3° lon × 2° lat"

- pg 7, 16-21 - Am I correctly interpreting that OMI NO$_2$ VCD in Figure 5 uses a AMF consistent with that emission inventory?

**Response:** We used the same satellite NO$_2$ products for comparison with all cases in the original manuscript. We have revised this issue and the OMI NO$_2$ VCD in Figure 5 uses AMF consistent with that emission inventory for now.

- pg 8, 27 - more details of how the counties are assigned would be useful.

**Response:** In this work, four municipalities are defined as a separate category. For the remaining cities, urban counties and suburban counties are divided according to the administrative district definition in China. We have revised the sentence as: "To further understand the biases in the model simulations, we divide all counties into three categories—counties in four municipalities (i.e., Beijing, Tianjin, Shanghai and Chongqing), urban counties, and suburban counties according to administrative district definition—and compare the modeled and observed $NO_2$ columns for these three categories."

- pg 10, 11 - Rephrase. It is not clear which error you are referring to.

**Response:** Revised as "The stated uncertainties of individual DOMINO v2.0 $NO_2$ column retrievals are estimated at $1.0 \times 10^{15}$ molecules $cm^{-2} + 25\%$".

- pg 11, 6-7 - This is most interesting for the S4 vs S5 differences. The use of newer road networks seems like an obvious best-practice. The IGDP seems like a less obvious improvement, which is why seeing the contribution of each would be nice.

**Response:** We have added a new scenario in the revised manuscript and concluded that improvements in the transportation sector have more significant effects on modeled $NO_2$ columns than that in the industrial sector.

Technical Corrections:
- pg 1, 26 - use IGDP here since that is how it is subsequently used or remove acronym, which is not used in the abstract or elsewhere without redefinition.

**Response:** Revised.

- Table 1 - citations in table should, but do not appear in bibliography.

**Response:** The citations are added in the bibliography.

- Fig 6 - How did your PDF bandwidth affect the shape of S1 and the tails of S2-S3?

**Response:** We use bandwidth of 0.1 in Figure 6, and we also test another two bandwidth, 0.05 and 0.2, as shown below. Curves with bandwidth of 0.05 is under smoothed since it contains too many spurious data artifacts. Curves with bandwidth of 0.2 is over smoothed and obscures much of the underlying structure. However, the selection of bandwidth will not affect our conclusion that S1 underestimates in some counties and S2 overestimates in some counties.

[Figure]

References:

Irie, H., Kanaya, Y., Akimoto, H., Tanimoto, H., Wang, Z., Gleason, J. F., and Bucsela, E. J.: Validation of OMI tropospheric NO$_2$ column data using MAX-DOAS measurements deep inside the North China Plain in June 2006: Mount Tai Experiment 2006, Atmos. Chem. Phys., 8, 6577-6586, 2008.

Kurokawa, J., Ohara, T., Morikawa, T., Hanayama, S., Janssens-Maenhout, G., Fukui, T., Kawashima, K., and Akimoto, H.: Emissions of air pollutants and greenhouse gases over Asian regions during 2000–2008: Regional Emission inventory in ASia (REAS) version 2, Atmos. Chem. Phys., 13, 11019-11058, 10.5194/acp-13-11019-2013, 2013.

Lamsal, L. N., Martin, R. V., van Donkelaar, A., Celarier, E. A., Bucsela, E. J., Boersma, K. F., Dirksen, R., Luo, C., and Wang, Y.: Indirect validation of tropospheric nitrogen dioxide retrieved from the OMI satellite instrument: Insight into the seasonal variation of nitrogen oxides at northern midlatitudes, J. Geophys. Res., 115, D05302, doi:10.1029/2009JD013351, 2010.

Lin, J.-T., R. V. Martin, K. F. Boersma, M. Sneep, P. Stammes, R. Spurr, P. Wang, M. Van Roozendael, K. Clémer, and H. Irie: Retrieving tropospheric nitrogen dioxide from the Ozone Monitoring Instrument: Effects of aerosols, surface reflectance anisotropy, and vertical profile of nitrogen dioxide, Atmos. Chem. Phys., 14, 1441-1461, doi:10.5194/acp-14-1441-2014, 2014.

Streets, D. G., Bond, T. C., Carmichael, G. R., Fernandes, S. D., Fu, Q., He, D., Klimont, Z., Nelson, S. M., Tsai, N. Y., Wang, M. Q., Woo, J.-H., and Yarber, K. F.: An inventory of gaseous and primary aerosol emissions in Asia in the year 2000, J. Geophys. Res., 108, 8809, doi:10.1029/2002JD003093, 2003.

Woo, J.-H., Baek, J. M., Kim, J.-W., Carmichael, G. R., Thongboonchoo, N., Kim, S. T., and An, J. H.: Development of a multi-resolution emission inventory and its impact on sulfur distribution for Northeast Asia, Water Air Soil Poll., 148, 259–278, 2003.

Zhang, Q., Streets, D. G., Carmichael, G. R., He, K. B., Huo, H., Kannari, A., Klimont, Z., Park, I. S., Reddy, S., Fu, J. S., Chen, D., Duan, L., Lei, Y., Wang, L. T., and Yao, Z. L.: Asian emissions in 2006 for the NASA INTEX-B mission, Atmos. Chem. Phys., 9, 5131–5153, doi:10.5194/acp-9-5131-2009, 2009.

Zheng, B., Huo, H., Zhang, Q., Yao, Z. L., Wang, X. T., Yang, X. F., Liu, H., and He, K. B.: High-resolution mapping of vehicle emissions in China in 2008, Atmos. Chem. Phys., 14, 9787-9805, 10.5194/acp-14-9787-2014, 2014.

---

## Author Response (AR2)

**Responses to editorial comments:**

Please address the remaining issue raised by Referee #1. Please also be sure that the MEIC web site is fully accessible and readable for English speakers before publication of this manuscript.
Once both of these issues are resolved, the manuscript will be ready for publication.

**Response:** The remaining issue from Referee #1 has been addressed. English version of the MEIC website (http://www.meicmodel.org) has been added.

**Responses to Anonymous Referee #1:**

The authors did a good job responding to requested updates except for one. In its current form, the article is acceptable to this reviewer for publication. However, I will *request* that the authors add language somewhere in the methods or concluding remarks about how S1 and S2 methodologies compare to typical applications.

**Response:** We appreciate that the referee is satisfied with our revisions. We also thank the referee for pointing out the remaining issue, which is addressed as below.

In my first review, I asked the authors to add comments about the relationship between S1/S2 and the other surrogate simulations. In their response to my review, the authors note that population density has been used for SMALL power plants, industry and "transportation." This does not address the fact that their "S1" uses population density for LARGE power plants, which I have not seen done elsewhere. Given that large power plants are a major NOx source, this should be noted as a bounding simplification. By including these bounding simplification scenarios (S1 and S2), the manuscript shows much larger sensitivity to spatial proxies than it would otherwise. For example, the $R^2$ range (max - min) for "All districts and counties" increases from 2 percentage points to 10 percentage points and for "Counties in municipalities" the $R^2$ range increases from 6 percentage points to 31 percentage points.
As the author's results show, the S1 and S2 results are primarily good justification for applying the more typical spatial allocation of large power plants. I am simply recommending that discussion be present to put the sensitivity shown in context.

**Response:** As the referee pointed out, S1 and S2 scenarios in this work, which used single proxy (i.e., population density/nighttime light) to allocate all emissions, are not common practice in current emissions inventories. We begin from the simplified case and compared it with S3 (locations for power sector and population density for other sectors) to emphasize the importance of knowing exact locations of large point sources in emission inventories, as

stated in the manuscript (in Sect. 3.1). We believe the main conclusions are robust because remarkable differences were found when comparing S4-S6 to S3 (which represents a common practice). For instance, $R^2$ increased from 0.87 in S3 to 1.01 in S6 for "All districts and counties" and increased from 0.67 in S3 to 1.13 in S6 for "urban districts".

To more clearly state the issue, we have added the following sentence in the Sect. 2.2 of the revised manuscript: "
[revised manuscript text omitted]